# Modernizing Neuro-Oncology: The Impact of Imaging, Liquid Biopsies, and AI on Diagnosis and Treatment

**DOI:** 10.3390/ijms26030917

**Published:** 2025-01-22

**Authors:** John Rafanan, Nabih Ghani, Sarah Kazemeini, Ahmed Nadeem-Tariq, Ryan Shih, Thomas A. Vida

**Affiliations:** Department of Medical Education, Kirk Kerkorian School of Medicine at UNLV, 625 Shadow Lane, Las Vegas, NV 89106, USA; rafanj1@unlv.nevada.edu (J.R.); ghani@unlv.nevada.edu (N.G.); kazems1@unlv.nevada.edu (S.K.); nadeemta@unlv.nevada.edu (A.N.-T.); shihr2@unlv.nevada.edu (R.S.)

**Keywords:** glioblastoma, radiomics, neuro-oncology, artificial intelligence, positron emission tomography, deep learning, convolutional neural networks, liquid biopsy, transfer learning, pseudoprogression

## Abstract

Advances in neuro-oncology have transformed the diagnosis and management of brain tumors, which are among the most challenging malignancies due to their high mortality rates and complex neurological effects. Despite advancements in surgery and chemoradiotherapy, the prognosis for glioblastoma multiforme (GBM) and brain metastases remains poor, underscoring the need for innovative diagnostic strategies. This review highlights recent advancements in imaging techniques, liquid biopsies, and artificial intelligence (AI) applications addressing current diagnostic challenges. Advanced imaging techniques, including diffusion tensor imaging (DTI) and magnetic resonance spectroscopy (MRS), improve the differentiation of tumor progression from treatment-related changes. Additionally, novel positron emission tomography (PET) radiotracers, such as ^18^F-fluoropivalate, ^18^F-fluoroethyltyrosine, and ^18^F-fluluciclovine, facilitate metabolic profiling of high-grade gliomas. Liquid biopsy, a minimally invasive technique, enables real-time monitoring of biomarkers such as circulating tumor DNA (ctDNA), extracellular vesicles (EVs), circulating tumor cells (CTCs), and tumor-educated platelets (TEPs), enhancing diagnostic precision. AI-driven algorithms, such as convolutional neural networks, integrate diagnostic tools to improve accuracy, reduce interobserver variability, and accelerate clinical decision-making. These innovations advance personalized neuro-oncological care, offering new opportunities to improve outcomes for patients with central nervous system tumors. We advocate for future research integrating these tools into clinical workflows, addressing accessibility challenges, and standardizing methodologies to ensure broad applicability in neuro-oncology.

## 1. Introduction

Brain tumors are rare malignancies that exhibit high mortality rates and cause severe neurological complications, including death. The 2021 World Health Organization (WHO) Classification of Tumors of the Central Nervous System (CNS) categorizes CNS tumors using histological findings, molecular biomarkers, and genetic profiles [1]. The most common malignant primary brain tumors in adults are diffuse gliomas, which are further categorized into astrocytoma, oligodendroglioma, and glioblastoma multiforme (GBM) (WHO grade IV). GBM accounted for 50.9% of all primary malignant tumors in the United States from 2016 to 2020 [2]. Despite current treatment strategies—maximal surgical resection followed by radiotherapy with adjuvant temozolomide (TMZ)—the prognosis for GBM is poor, with a median survival time of 15–16 months and a 5-year survival rate of 5–10% [3,4,5].

Brain metastases carry a much worse prognosis than primary brain tumors, with the average survival time typically being less than six months [6]. Brain metastases occur ten times more frequently than primary brain tumors, commonly originating from the lungs, breast, and skin [7,8,9,10]. Brain metastasis poses a significant treatment challenge primarily due to therapeutics being unable to cross the blood-brain barrier (BBB) [11]. Additionally, the development of micrometastases, which are undetectable metastatic lesions, contributes to the poor patient outcomes seen in metastatic brain tumors [11]. Brain tumors have various risk factors, with the most established one being ionizing radiation; genetic predisposition also plays a role in brain tumor development [12].

These statistics for CNS tumors, primary and secondary, reflect the need for novel tools in diagnosing both primary and metastatic brain tumors. Brain metastasis and GBM are complex tumors to diagnose, but early diagnosis and treatment can lead to better patient outcomes. Early surgical intervention extends postoperative survival in brain cancer patients to 28.4 months, compared to 18.7 months for delayed surgery [13]. Advancements in diagnostic tools, such as advanced magnetic imaging, novel positron emission tomography (PET) radiotracers targeting amino acids and short-chain fatty acids, artificial intelligence (AI), and liquid biopsies, revolutionize neuro-oncology. These innovations overcome critical diagnostic barriers, including the differentiation of pseudoprogression from actual tumor progression, the detection of radiation necrosis, and the precise identification of tumor recurrence or metastasis.

We hypothesize that integrating these novel tools with existing diagnostic frameworks will enhance early detection, refine treatment strategies, and improve patient outcomes. This critical review examines how these technologies advance neuro-oncology diagnostics and shape the future of personalized medicine.

## 2. Challenges in Neuro-Oncology Imaging: Addressing the Limitations of MRI

Initial evaluation for primary and metastatic brain tumors in adult and pediatric patients remains magnetic resonance imaging (MRI) with contrast, allowing for visualization of anatomical structures. MRI characterizes brain tumors, therapeutic response, recurrence, and postoperative status [14,15,16]. The MRI’s increasing global availability, lack of radiation exposure, higher resolutions, and 3D analysis make it practical in clinical environments [17]. However, with current standard MRI imaging, limitations and challenges exist, including instances where tumor visualization may be difficult and complex. In this section, we will discuss MRI’s limitation in differentiating proper tumor growth from pseudoprogression and tumor necrosis, and the presence of a BBB restricting the movement of the MRI’s most common contrast agent, gadolinium (Figure 1).

This emphasizes the diagnostic challenges associated with pseudoprogression, radiation necrosis, and BBB-related limitations, highlighting the need for advanced imaging and diagnostic techniques to improve post-therapy monitoring.

### 2.1. Gadolinium-Based Agents: Challenges in Crossing the Blood-Brain Barrier

In pathological conditions such as brain tumors, where non-contrast MRI imaging alone cannot visualize the mass, a contrast dye, gadolinium, may be injected for more accurate visualization. Gadolinium-based agents are widely used in contrast imaging due to their paramagnetic properties, which arise from their seven unpaired electrons—the maximum for any ion. This property significantly enhances MRI, making gadolinium a mainstay in medical imaging [18]. Additionally, gadolinium continues to be used due to its mild safety profile. Ninety percent of this contrast is excreted into the urine in patients with normal kidney function, and the side effect profile includes less severe symptoms such as headache, nausea, and pain at the injection site [19].

One concern with gadolinium is that it may deposit in the brain upon repeated use, causing neurotoxicity [20]. It is well described that patients with severe kidney disease can develop nephrogenic systemic fibrosis due to increased levels of circulating free gadolinium because of poor excretion [18]. The circulating gadolinium can cause fibrosis throughout the body (bone and kidneys). However, the effect of gadolinium deposition has not been sufficiently studied to evaluate neurological sequelae and continues to be an area of research [21,22,23]. Gadolinium deposition is uncommon as it can be chelated to a carrier protein, reducing off-target effects and preventing the toxic accumulation of free gadolinium in areas such as the brain [18,20]. Despite potential deposition, gadolinium is continuously used because of its relatively safe side effect profile and low occurrence of off-target effects due to chelators.

The BBB poses a significant challenge in the administration of MRI contrast. The BBB maintains brain homeostasis by regulating nutrient passage and preventing toxic accumulation. Composed of microvascular endothelial cells, astrocytes, and pericytes, it restricts most substances except small lipid-soluble molecules and gases like O_2_ [24,25]. Ions and other substances essential for normal brain physiology require specific transporters, such as ATP-binding cassette (ABC) transporters. ABC transporters also mediate the efflux of neurotoxic agents and drugs, making it challenging to use radiocontrast or therapeutic agents within the brain unless the BBB is disrupted [26].

The growth of brain tumors can physically disrupt the established structure of the BBB, allowing the contrast agent, which would usually be kept out, to leak into the surrounding tissue, leading to increased intensity and enhancement on MRI [27]. The degree of BBB disruption affects the visualization of contrast on MRI. Although this is useful for seeing the extent of the brain tumor, it is also a limitation affecting accuracy and interpretation. Tumors such as GBM may not entirely disrupt the BBB and may contain regions of an intact BBB, which obscures the true extent of involvement. Due to these limitations, MRI shows only gross disruption and fails to capture fine details of different tumor regions [28]. Additional advanced techniques may be required to determine the extent of the tumor. Accordingly, many advances seen in imaging tools in neuro-oncology are directed toward improving the diagnosis of brain tumors with minimally invasive modalities that may not require BBB disruption.

### 2.2. Pseudoprogression: A Diagnostic Challenge in Post-Treatment Imaging

A limitation of conventional contrast-enhanced MRI is its inability to accurately distinguish pseudoprogression from actual tumor progression. Patients undergoing standard treatment, a combination of radiotherapy and TMZ, may experience transient deterioration and contrast enhancement changes on MRI [29,30,31,32]. This phenomenon is termed pseudoprogression, which is defined as a new area of enhancement on MRI post-radiation therapy (e.g., within 3–4 months) without true tumor growth and eventually subsiding without a change in treatment [31]. The incidence of pseudoprogression varies widely, affecting up to 32% of patients who show early progression [33]. Specific studies have reported incidences of 19.4% at three months post-radiation and 7.0% at six months post-radiation in GBM patients [34]. In patients with isocitrate dehydrogenase (IDH) mutant gliomas, post-surgery contrast enhancement occurred in 56% of cases, of which 28% were pseudoprogression and 55% were actual tumor progression [35]. Pseudoprogression can mask actual tumor progression and thus delay treatment options. The current literature does not adequately describe pseudoprogression pathophysiology, revealing gaps. The most widely accepted hypothesis suggests that radiation therapy increases local inflammation, resulting in abnormal permeability of the BBB and increased edema [36,37,38].

While MRI is the primary imaging modality in monitoring brain tumors preoperatively and postoperatively, pseudoprogression resembles actual tumor growth, complicating interpretation and potentially delaying treatment [31]. Pseudoprogression complicates tumor diagnosis, often leading to misinterpretation. A French analysis of high-grade gliomas treated with radiotherapy, conducted through the POLA network—a program managing de novo adult high-grade gliomas from January 2008 to January 2020—examined rates of pseudoprogression misdiagnosis [39]. The analysis identified retrospective misdiagnosis rates of 17% in both the reference and control groups for patients with true tumor progression [39]. Diagnosing true tumor progression and pseudoprogression is important because misdiagnosis can lead to different treatment options and delay appropriate treatment. Treatment for pseudoprogression is self-resolving, as the condition is transient; however, if tumor progression has occurred, more aggressive management is required: radiation, clinical trials, and/or surgery [29]. Therefore, MRI alone to delineate treatment-related changes may be inadequate and require more advanced techniques.

### 2.3. Disentangling Radiation Necrosis from Tumor Recurrence

Radiation necrosis is a potential complication from radiotherapy that can lead to an array of severe neurological complications, including acute encephalopathy, focal cerebral necrosis, and cerebrovascular disease [40]. Radiation necrosis is a potential consequence of radiation therapy occurring several months post-treatment (e.g., 1–2 years) [41,42]. A significant limitation of conventional contrast-enhanced MRI is accurately detecting tumor radiation necrosis and differentiating it from pseudoprogression and true tumor progression [43]. This is mainly due to the resulting edema, cell necrosis, and vascular permeability, which obscure MRI readings, affecting diagnosis and patient survival [44].

The pathophysiology of radiation necrosis arises from several processes when the brain is exposed to radiation. Radiation exposure induces reactive oxygen species that damage DNA and increase cytokine production [44]. Additionally, radiation can interact with the cytoplasmic membrane, causing ceramide-induced apoptosis, leading to increased cell swelling and cell necrosis of endothelial cells [45]. This endothelial damage further perpetuates BBB disruption, fibrin-platelet thrombosis, and fibrinoid necrosis. Moreover, radiation damages white matter, affecting oligodendrocytes, astrocytes, and neural progenitor cells. Dysfunctional astrocytes upregulate vascular endothelial growth factor (VEGF) and hypoxia-inducible factor-1α (HIF-1α). HIF-1α induces VEGF, leading to angiogenesis and increased vascular permeability. Dysfunctional oligodendrocytes result in demyelination [44,45]. Radiation necrosis alters MRI readings, necessitating additional diagnostic testing.

The incidence of radiation necrosis depends on the radiation dose, where higher radiation levels indicate increased risk [46]. While both pseudoprogression and radiation necrosis occur as complications from radiation therapy, pseudoprogression is an acute inflammatory response to radiation treatment occurring in a shorter time frame, within 3 months, and radiation necrosis is a delayed injury occurring months after therapy [44]. Accurate differentiation of pseudoprogression, radiation necrosis, and true tumor progression is essential for diagnosis and directing treatment options. Radiation necrosis, a more severe complication than pseudoprogression, often requires treatment. Options include glucocorticoids, anticoagulants, and anti-VEGF therapies like bevacizumab, nerve growth factor, anti-inflammatory injections, surgery, hyperbaric oxygen, and laser thermal ablation [45]. Despite current advances in diagnostic imaging, standard MRI may not accurately detect tumor necrosis. More novel techniques delineate these complications, aiding in improved diagnostic capabilities.

Within the past decade, significant advancements in novel diagnostic tools have addressed the challenges and limitations of conventional MRI: pseudoprogression, radiation necrosis, and gadolinium crossing a highly selective BBB. Novel diagnostic tools classify CNS tumors with higher accuracy and precision. Significant improvements in diagnostic imaging modalities are seen in advanced MRI imaging, leading to earlier personalized and targeted therapies.

## 3. Transforming Neuro-Oncology Diagnostics Through Advanced MRI Techniques

Conventional MRI is the primary imaging modality for brain tumors because it specifies the location and extent of involvement of brain masses [47]. Due to the limitations of traditional MRI, more advanced and improved MRI techniques have been developed to aid in diagnostic accuracy and prevent misdiagnosis. Traditional MRIs are exceptional in diagnosing specific tumor types; however, some tumor types, especially gliomas, can be confused with metastasis, meningiomas with solitary fibrous tumors, and pituitary adenomas with craniopharyngiomas [48]. A case-control study evaluated the key features of diffuse gliomas that were misinterpreted on initial imaging [49]. The analysis identified 18 missed gliomas, with six attributed to perceptual errors where the glioma was visible but not recognized on neuroimaging. The remaining twelve cases were classified as interpretive errors, where the glioma was identified but misdiagnosed as a different pathological condition. Missed cases were typically smaller, less likely to enhance after imaging, and exhibited minimal mass effect [49]. Conventional MRI diagnostic accuracy varies due to measurement error, observer variability, and confounding factors like pseudoprogression or radiation necrosis. Advanced MRI techniques offer a promising solution to these diagnostic challenges.

### 3.1. Mapping Brain Tumor Invasion: The Role of Diffusion Tensor Imaging (DTI)

Advanced MRI techniques distinguish between pseudoprogression and actual tumor progression, as early detection is essential in preventing devastating neurological consequences. Advanced MRI techniques offer more detailed insight into biochemical pathways, tissue compositions, and metabolic and physiological processes than conventional MRI [50]. Standard advanced MRI techniques already seen in practice include T1-weighted imaging, T2-weighted imaging, and T2-fluid attenuated inversion recovery (FLAIR). More recent advanced MRI techniques that are being extensively studied for clinical use in diagnosing brain cancers include magnetic resonance spectroscopy (MRS), diffusion tensor imaging (DTI), functional MRI, diffusion-weighted imaging (DWI), susceptibility-weighted imaging (SWI), MR chemical exchange saturation transfer (CEST), and MRI perfusion [51].

This section will focus on the recent advances with MRS and DTI. These diagnostic tools represent some of the more extensively studied modalities in characterizing tumors and differentiating actual tumor progression from pseudoprogression and radiation necrosis. Both advanced MRI diagnostic tools provide more specific details about brain tumors and a greater understanding of disease mechanisms in a non-invasive manner. While standard MRI provides structural and gross information, advanced MRI diagnostic tools provide more comprehensive and functional information, making them useful in pre- and post-operative domains [52]. Modern diagnostic techniques have transitioned to include metabolic and morphological characteristics to facilitate more precise tumor management.

DTI is frequently used in surgical planning for basic white matter tract visualization. DTI helps identify any disruptions in white matter tracts resulting from tumors or inflammatory processes [51,53,54]. This technology has been used in various demyelinating diseases such as multiple sclerosis, Alzheimer’s, and amyotrophic lateral sclerosis [55]. In the context of brain tumors, it is used for surgical planning. Other imaging modalities include high angular resolution diffusion imaging (HARDI) and diffusion spectrum imaging (DSI), which have potential use when additional information, like the visualization of crossing or more complicated white matter tracts, is required [56]. However, these have been studied much less in the application of surgical planning for brain tumors and are significantly more expensive. DTI is a promising advanced neuro-oncology MRI technique that improves brain tumor assessment, diagnosis, and treatment.

Through visualization of white matter tracts, DTI visualizes brain structures by measuring the diffusion of water molecules. Water molecules move freely in all three spatial directions in the body, but diffusion is constrained along specific directions in organized structures like the brain. For example, white matter tracts diffuse along the direction of the fibers rather than across them. This anisotropic diffusion is key to visualizing the brain structures [57]. Disruptions to this diffusion pattern, such as those caused by brain tumors, alter key DTI metrics, such as fractional anisotropy (FA). FA values provide information about the degree of directional diffusion, and changes in FA indicate underlying pathology. A change in FA less than −30% indicates white matter disruption; a positive FA change suggests edema or displacement; an FA change value between 0–30% typically signals tissue displacement or infiltration [58].

The FA metric is linked to varying degrees of white matter disruption, offering a quantifiable method for assessing tumor impact. By highlighting FA and their corresponding interpretations, DTI can provide evidence-based conclusions in clinical practice. Assessing white matter involvement helps determine brain tumor behavior accurately, further supporting the advantages of DTI. Displacement of white matter tracts suggests a benign course, while more malignant tumors are associated with the destruction and infiltration of white matter tracts. In a study analyzing the effects of tumors on white matter tracts, DTI identified 40% of cases involving infiltration, 76.7% as displacement, and 20% as destruction [59]. Other metrics, such as axial diffusivity, planar tensor, and spherical tensor, displayed AUC greater than 0.5 and were statistically significant in distinguishing high-grade gliomas from low-grade gliomas [60]. Integrating these quantitative metrics provides insights into assessing tumor impact on white matter tracts and helps guide clinical decision-making.

A systematic review analyzed DTI in tumor resection surgery, finding that this technique led to more significant removal of the tumor and a lower chance of postoperative impairment compared to the non-DTI group [61]. Additionally, surgeries utilizing DTI had better patient outcomes pre-operatively as it could predict postoperative deficits and increase the probability of preserving brain function. Utilizing DTI to guide tumor resection aids in better surgical planning because more of the tumor is visualized appropriately [57]. DTI in awake brain surgeries reduced operative time by an average of 26.2 min compared to surgeries without DTI [62]. The analysis also compared sensitivity and specificity for complete tumor resection. Surgeries without DTI achieved a sensitivity of 88% and a specificity of 62.5%, while those with DTI showed improved sensitivity of 100% and specificity of 80% [62]. These operative time, sensitivity, and specificity improvements are critical for reducing neurological complications and enhancing patient outcomes.

Current work with DTI is better aimed at identifying brain tumors despite edema. One limitation of using DTI is locating tumors in the presence of edema because it can obscure accurate visualization of the white fiber tracts and, thus, affect surgical planning [62]. An analysis method using a free water model replicates the contribution of edema separately from white fiber tracts [63]. This model tracks white fiber tracts through the arcuate fasciculus in regions of edema. Incorporating the free water model alongside tractography improved white matter monitoring in areas of edema. Including this approach increases the sensitivity of white matter tracts, enhancing surgical planning in affected regions [63].

DTI has emerged as a key tool for detecting treatment-related changes. In GBM patients, DTI improves diagnostic accuracy when evaluating treatment response and pseudoprogression [64]. Additionally, DTI helps predict glioma recurrence, enabling better tumor management [65,66].

A novel advancement in DTI is incorporating artificial intelligence (AI) to enhance diagnostic capabilities. Although rarely described, one approach combined a convolutional neural network, structural MRI, and DTI to predict IDH mutation status in glioma patients [67]. This method analyzed DTI metrics, such as mean diffusivity and fractional anisotropy, alongside MRI metrics, including fluid-attenuated inversion recovery, non-enhanced T1, and T2-weighted images—parameters known to predict IDH status. Using an AI model, the analysis extracted these metrics to predict IDH mutation status, achieving sensitivity and specificity of 0.925 and 0.567, respectively, outperforming imaging modalities alone [67]. Integrating DTI with a machine learning model also improved IDH status prediction accuracy, yielding an AUC of 0.900 (95% CI, 0.855–0.945; *p* = 0.006) in glioma patients [68]. Section 5 below discusses in greater detail the transformative role of AI in neuro-oncology diagnostics.

White matter fiber tracking in DTI ultimately mitigates maximal tumor resection and increases survival [62,69]. Its application in surgical planning visualizes the extent of involvement of invasive tumors that MRI cannot reveal, yielding a more definite diagnosis and thus improving patient outcomes. The improvements in DTI, free water models, and AI have improved white matter tract visualization. Recent studies explore DTI’s use outside of surgical planning and more as a diagnostic tool, such as detecting IDH mutation status in gliomas. This adds to DTI’s value in improving patient outcomes from a surgical and diagnostic perspective. Future work in DTI should continue to make the diagnostic process more efficient and accurate.

### 3.2. Metabolic Profiling with Magnetic Resonance Spectroscopy (MRS): Insights into Tumor Type, Grade, and Recurrence

More advanced imaging, such as MRS, discerns the differences between brain tumors that may present complex findings and challenges. Accurate and early detection of complex brain tumors prevents severe neurological conditions and improves patient outcomes. MRS is a diagnostic modality that accurately identifies brain tumors through metabolic levels [70]. Specifically, MRS has proven helpful in delineating between tumor progression, recurrence, and radiation necrosis [71]. A meta-analysis observed improved MR perfusion and MRS accuracy in 13 studies in differentiating necrosis from recurrent tumors in measuring metabolite levels [72]. This analysis concluded that measuring the ratios of unique metabolites in tumor cells may improve diagnostic accuracy [72]. Altered metabolism is a distinct feature of tumorigenesis, and MRS is a unique diagnostic tool capable of detecting abnormal metabolites in tumor tissues [73]. MRS quantifies specific isotopes such as ^1^H, ^13^C, and ^31^P in brain tissues, whose levels are often altered in pathological states [74]. Significant improvements continue to occur using MRS in classifying and understanding the pathophysiology of brain tumors related to metabolite levels.

One clinical application of MRS is determining tumor recurrence, type, and grade. MRS measures the concentrations of various metabolites within tumor tissues, such as amino acids, lipids, and neurotransmitters. These metabolites provide insights into the brain tumor environment. Common substances assessed on MRS include creatine, N-acetyl aspartate, choline, lactate, and lipids [75]. Elevated choline indicates cell proliferation and thus indicates tumorigenesis [76]. Similarly, abnormally elevated levels of N-acetyl aspartate in the brain suggest the presence of a brain tumor [77,78,79]. Other metabolite levels, such as lactate and lipids, may represent the metabolic change in higher-grade tumors, revealing anaerobic glycolysis use and adaptation to hypoxic conditions [75]. Various studies explore the associated MRS ratios and cutoffs in patients with treatment-related changes [80]. Increased lipid levels and low choline/N-acetyl aspartate aid in differentiating pseudoprogression from actual progression, while increased choline/creatine is observed in radiation necrosis [80]. These abnormal levels of N-acetyl aspartate, choline, lipids, and lactate help detect brain tumors.

Since the development of MRS, significant data analysis and hardware advancements have improved its diagnostic capabilities [81]. Integrating AI further enhances MRS and improves the interpretation of complex metabolic data from brain tumors. Differentiating high-grade gliomas from low-grade ones remains challenging, as both exhibit abnormal metabolic levels. Incorporating AI into MRS analysis identifies key metabolic differences, achieving an accuracy of 87% in classifying high-grade and low-grade gliomas [82]. Additionally, MRS and machine learning methods have been used to evaluate high-grade glioma recurrence [83]. By analyzing tumor metabolites in post-treatment patients, machine-learning models predicted tumor progression with an AUC of 0.86, enabling prediction of tumor recurrence up to eight months in advance [83].

MRS is a unique diagnostic tool in neuro-oncology. It uses metabolites to diagnose brain tumors. Future work with MRS should explore AI and its ability to interpret metabolic values. The incorporation of AI in MRS represents a diagnostic advancement that improves the accuracy of detecting early tumor progression, differentiating treatment-related changes, and distinguishing high-grade gliomas from low-grade gliomas.

## 4. Advancing Neuro-Oncology Diagnostics: The Role of PET Imaging and Emerging Radiotracers

PET imaging is a non-invasive nuclear medical imaging modality that combines functional and structural information. It is crucial in assessing tumor response to therapy, detecting and staging tumors, and monitoring recurrence [84]. PET imaging typically requires the use of different types of positron-emitting radiopharmaceuticals, such as ^11^C-methionine (^11^C-MET), ^18^F-fluoroethyltyrosine (^18^F-FET), phenylalanine (^18^F-DOPA), and ^11^C-alpha-methyl-L-tryptophan (^11^C-AMT). However, the most clinically used radiotracer remains ^18^F-fluorodeoxyglucose (^18^F-FDG), a glucose analog [85].

Current guidelines from the European Association of Nuclear Medicine (EANM), European Association of Neuro-Oncology (EANO), Society of Nuclear Medicine and Molecular Imaging (SNMMI), and Response Assessment in Neuro-Oncology (RANO) outline clinical scenarios for PET imaging use [86]. Circumstances where PET scans are helpful include malignant differentiation of higher-grade gliomas, delineating pseudoprogression from actual progression, and monitoring recurrence [86]. More importantly, PET scans can be utilized when MRIs prove nondiagnostic, and more extensive imaging may be required. However, general practice continues to use PET/CT scans and the glucose radiotracer because these tools are more extensively studied and widely available.

Radiotracers, such as ^18^F-FDG and ^11^C-MET, offer improved tumor visualization and metabolic profiling. While ^18^F-FDG highlights tumors with elevated glycolysis, ^11^C-MET provides superior tumor-to-background contrast [87]. Radiotracers possess several key characteristics: lipophilicity, low molecular weight, no formal charge, and minimal adverse effects [88,89]. Crossing an intact BBB and selectively reaching its target tumor [87] is the most crucial characteristic. Specific radiotracers have unique properties that explore different biochemical pathways, providing a deeper understanding of brain tumors.

Radiotracer metrics, standard uptake value (SUV), time to peak (TTP), and tumor-to-background ratio (TBR) are essential values in diagnostic imaging, providing insights into tumor characteristics and behavior. The SUV quantifies the concentration of a radiotracer in a specific region, reflecting its absorption and metabolic activity. When utilizing ^18^F-FDG, an SUVmax/WM of 1.90 is the cutoff for predicting maximal survival post-treatment in primary brain tumor patients [90]. TTP is defined as the interval from radiotracer injection to its maximum uptake, with longer TTP values associated with a favorable prognosis and shorter TTP values observed in aggressive tumors [91]. ^18^F-FDG is helpful for initial imaging; however, due to the presence of GLUT-3 transporters in normal brain tissue, it can be challenging to diagnose low-grade tumors [91,92]. Utilizing other radiotracers, such as ^18^F-FET, and measuring TBR is critical in distinguishing malignant activity from surrounding non-cancerous brain tissue. A TBR mean of 2.0 and TTP greater than 45 min identified glioma recurrence with a sensitivity of 93% and specificity of 100% when using ^18^F-FET [93]. Elevated TBR is linked to poorer survival outcomes, underscoring its prognostic importance [92]. Quantifiable metrics such as SUV, TTP, and TBR provide a framework for distinguishing tumor types and assessing recurrence, advancing evidence-based neuro-oncological care.

This section will discuss innovative diagnostic tools such as ^18^F-fluoropivalate (^18^F-FPIA), ^18^F-FET, and ^18^F-fluciclovine (^18^F-FACBC) and the hybrid use of PET/MRI. Table 1 summarizes the structural analog, proposed use, and clinical significance of ^18^F-FDG, ^18^F-FPIA, ^18^F-FET, and ^18^F-FACBC. We will evaluate ^18^F-FPIA, ^18^F-FET, and ^18^F-FACBC to assess tumor type and grade and detect malignancy.

### 4.1. ^18^F-FPIA: Unlocking the Diagnostic Potential of Fatty Acid Metabolism in High-Grade Gliomas

Radiotracers provide valuable metabolic insights as diagnostic tools for CNS tumors. While increased ^18^F-FDG uptake is typical in tumor cells because of increased glucose metabolism, ^18^F-FDG is not sensitive in differentiating lower-grade tumor cells or detecting pseudoprogression [85]. Not all brain tumors display increased utilization of glucose. Higher-grade tumors adjust to their microenvironments, driving fatty acid oxidation instead [94,95,96]. This unique metabolism present in higher-grade gliomas is used as a method for detection. Higher-grade gliomas rely on fatty acids more than carbohydrates due to fatty acids providing more ATP, thus playing a critical role in the tumorigenesis of gliomas [97,98]. Therefore, radiotracers that exploit fatty acid metabolism are future directions for detecting brain tumors.

Among developing radiotracers, ^18^F-FPIA, a short-chain fatty acid analog, offers insights into fatty acid metabolism in higher-grade gliomas [99,100]. An analysis of 10 adult glioma patients with varying tumor grades measured ^18^F-FPIA uptake. Results showed that higher-grade gliomas (grade IV) exhibited greater radiotracer uptake than lower-grade gliomas (grade II-III), although the exact mechanism remains unknown [101]. The safety profile of ^18^F-FPIA has been favorable, with no adverse effects or changes in vital signs, laboratory values, or electrocardiograms reported during or 24 h post-injection [102].

While the short-chain fatty acid radiotracer ^18^F-FPIA helps detect different grades of gliomas, evidence supporting its efficacy in predicting therapy-related complications remains limited. The ongoing clinical trial, ^18^F-FPIA PET-CT in Glioblastoma Multiforme (GBM) (FAM-GBM) (NCT05801159) [103], quantifies fatty acid oxidation to evaluate its use in postoperative GBM patients undergoing chemoradiotherapy [104]. This trial evaluates radiotracer uptake in GBM patients after treatment and explores its potential diagnostic value in detecting higher-grade tumors and discerning post-treatment changes. However, the study’s small sample size and focus solely on higher-grade gliomas limit its generalizability. Future clinical trials should assess ^18^F-FPIA uptake in higher- and lower-grade gliomas to determine whether it can reliably differentiate tumor grades. Expanding sample sizes would also help validate these findings. When appropriately applied, ^18^F-FPIA PET could provide valuable insights into treatment-related changes, particularly in cases where other diagnostic modalities fall short.

### 4.2. Decoding Tumor Complexity with an Amino Acid Radiotracer: ^18^F-Fluoroethyltyrosine (^18^F-FET)

Similarly, radiotracers can identify brain tumors based on amino acid metabolism, enhancing our understanding of brain tumor physiology. PET scans utilizing amino acid radiotracers may be superior to advanced MRI techniques. For one, tumor cells have increased System L amino acid transporters (*LAT1* and *LAT2*) [105]. This characteristic allows amino acid radiotracers to pass the BBB and achieve better tumor-to-background contrast than MRI [93]. Among the various studied amino acid radiotracers, ^18^F-FPIA may help discern treatment-related changes.

^18^F-FET can differentiate between tumor progression and responsiveness to treatment. Compared to standard MRI imaging, using ^18^F-FET as a radiotracer in PET scans aids in differentiating tumor recurrence from treatment-related changes [106,107,108]. MRI imaging and ^18^F-FET PET/CT as separate modalities to differentiate true tumor progression and radiation necrosis had lower sensitivity, specificity, positive predictive value (PPV), and negative predictive value (NPV). A combined MRI and ^18^F-FET PET/CT yielded better sensitivity, specificity, PPV, and NPV results: 97.9%, 100%, 100%, and 91.6%, respectively [107]. A hybrid modality combining ^18^F-FET perfusion PET/MRI demonstrated an overall accuracy of 86%, significantly higher than the 66% accuracy achieved with standard MRI in complex cases, including treatment-related changes [109].

Moreover, ^18^F-FET can identify new malignancies and distinguish treatment-related changes. ^18^F-FET PET/MRI identifies true tumor progression in patients undergoing therapy and detects new malignancies in untreated lesions with an accuracy of 85% [110]. These findings demonstrate the amino acid tracer’s ability to differentiate treatment-related changes from malignancy [110]. Additionally, ^18^F-FET may aid in detecting treatment-related changes from brain metastases [111].

A completed clinical trial evaluates the use of ^18^F-FET for assessing treatment-related changes in GBM patients (NCT01756352) [112]. This interventional study measures progression-free survival with ^18^F-FET in recurrent GBM patients receiving the anti-VEGF drug Avastin (bevacizumab). Avastin is commonly utilized for GBM patients failing first-line treatment, but patients may experience severe complications such as internal bleeding, renal toxicity, and severe hypertension. Predicting outcomes with ^18^F-FET PET allows adjustments to treatment options and prevents unnecessary complications. A limitation of the clinical trial is that it only evaluates Avastin-treated patients. Future studies should expand the use of ^18^F-FET to identify other treatment-related changes, such as pseudoprogression and radiation necrosis.

### 4.3. Sharper Contrasts, Clearer Answers: ^18^F-Fluciclovine (^18^F-FACBC) for Differentiating Tumor Grades

The amino acid radiotracer ^18^F-FACBC has the potential to identify brain tumor progression and differentiate tumor changes. A pilot analysis reported the accuracy of combined ^18^F-FACBC PET/MRI in distinguishing low-grade from high-grade gliomas. Higher-grade gliomas exhibit greater tumor-to-background contrast with ^18^F-FACBC compared to lower-grade gliomas [113]. Using this characteristic, sensitivity for tumor detection was analyzed across PET, contrast MRI, and combined PET/MRI. The combined modality achieved 100% sensitivity (*p* = 0.031), while PET alone reached 54.5% and contrast MRI alone at 45.5% [113]. Evidence suggests that ^18^F-FACBC may outperform extensively studied radiotracers such as ^11^C-MET due to its higher contrast resolution and longer half-life [114,115]. Ongoing and current research should validate the utility of ^18^F-FACBC in complex cases, such as distinguishing pseudoprogression from true progression [116]. Three ongoing clinical trials assess the efficacy of ^18^F-FACBC in brain tumors, determining whether this radiotracer provides greater accuracy in detecting treatment-related changes compared to existing diagnostic methods.

A Phase II clinical trial is investigating the efficacy of ^18^F-FACBC in detecting brain tumors and guiding surgical intervention for brain metastases and stereotactic radiosurgery (NCT05554302) [117]. This trial evaluates the use of ^18^F-FACBC in a hybrid PET/CT imaging modality to plan surgical intervention and radiosurgery. It will enroll 20 patients to assess whether the radiotracer accurately evaluates the extent of surgical resection and detects tumor recurrence. Additionally, the trial will determine if ^18^F-FACBC provides superior visibility compared to traditional MRI. This novel radiotracer enhances treatment planning through assessing treatment response following surgery or radiotherapy.

A Phase III clinical trial (NCT06159335) is exploring the accuracy of ^18^F-FACBC in assessing brain tumors in patients with brain metastases undergoing immunotherapy [103]. This trial will analyze 30 patients using an ^18^F-FACBC PET/MRI modality to quantify amino acid uptake in brain tumors. Higher ^18^F-FACBC uptake is hypothesized to indicate malignancy. If validated, ^18^F-FACBC could become a valuable diagnostic tool for characterizing and detecting brain tumors, potentially improving physicians’ clinical decision-making.

A third clinical trial is an observational Phase I study focused on the efficacy of ^18^F-FACBC in recognizing treatment-related changes, specifically radiation necrosis versus true tumor progression (NCT04462419) [118]. This ongoing study, which is not recruiting, evaluates static fluciclovine PET standardized uptake values (SUV peak and SUV mean) and metabolic tumor volumes in 19 patients using PET/MRI imaging. The trial will determine whether ^18^F-FACBC reliably differentiates true progression from radiation necrosis in patients with brain metastases. Findings from this study may refine and standardize PET imaging criteria for distinguishing tumor progression from radiation necrosis, advancing diagnostic precision in neuro-oncology.

Although these studies are under different phases and have yet to be completed, they may spark further research refining the use of ^18^F-FACBC in neuro-oncology. A current limitation in these clinical trials is the lack of a large sample size, with 30 participants being the most significant number of participants for one of the clinical trials. Some other limitations of these clinical trials include a narrowed focus on radiation necrosis. Expanding studies should consist of much larger diverse population sizes, evaluate treatment-related changes, and focus on criteria differentiating actual tumor progression from both radiation necrosis and pseudoprogression.

More research and clinical trials should focus on ^18^F-FACBC’s safety and clinical utility. In doing so, ^18^F-FACBC could positively impact patient management, providing more timely detection of tumor progression and radiation necrosis, leading to earlier treatment adjustments and intervention. A current summary of clinical trials surrounding ^18^F-FPIA, ^18^F-FET, and ^18^F-FACBC and the measured outcomes are listed in Table 2. Integrating these novel radiotracers beyond clinical trials holds promise for advancing neuro-oncology and improving diagnostic capabilities and outcomes for patients with brain metastasis and treatment-related changes.

### 4.4. Beyond the Scan: Limitations of PET Radiotracers and Advanced MRI

Limitations in PET radiotracers and advanced MRI techniques span various areas, including availability, training, and outdated software. These technologies are often challenging to access in rural areas because specialized software and machinery are typically only available in larger diagnostic centers. For developing regions, such as the African continent, the distribution of PET scans remains low [119]. Additionally, supervised training and expertise may be required to operate these sophisticated scans, exacerbating a lack of access in underdeveloped areas. Concerns exist regarding the production and distribution of radiotracers, especially in these regions. However, a study suggests that the availability of radiotracers can be made feasible through radionuclide generators [120].

Standardization for novel radiotracers remains challenging as metrics and cutoff values, such as SUV, have not been uniformly defined, leading to variability. While the safety and long-term effects of ^18^F-FDG have been extensively explored, there are scarce studies surrounding the safety of novel radiotracers: ^18^F-FPIA, ^18^F-FET, and ^18^F-FACBC. Ongoing and future research must establish cutoff values, such as knowing what SUV value ensures consistent evaluation of brain tumors. Some clinical trials in Table 2 aim to measure radiotracer uptake in brain tumors. Once cutoff values have been established, guidelines and appropriate utilization of radiotracers should be validated. ^18^F-FPIA has promise in detecting malignant brain tumor cases characterized by fatty acid metabolism and should be utilized here. ^18^F-FET and ^18^F-FACBC should be used to distinguish treatment-related changes. The versatility of existing PET scans in utilizing different types of radiotracers helps standardization, avoiding costly upgrades or additional software.

Standardizing advanced MRI techniques presents similar challenges, including high costs, susceptibility to artifacts, a lack of reference imaging data, and longer processing times [57,80,121,122]. However, this is addressed through targeted training, software upgrades, and AI integration. Advanced MRI techniques, like DTI and MRS, are extensions of MRIs that only require software, offering a pathway for standardization without costly efforts or the need to replace machines. Ensuring accessibility and consistency in implementing these novel machines will reduce institutional variations. More studies comparing the efficiency of advanced MRI techniques with existing diagnostic tools will aid in developing guidelines. Additionally, consensus among institutions regarding protocols and algorithms for when and how to conduct these advanced techniques could assist in standardizing them. AI integration can supplement these tools by reducing bias, mitigating variability, and expediting scan processing times.

### 4.5. Combining Strengths: PET/MRI for Enhanced Neuro-Oncology Diagnostics

Hybrid utilization of PET/MRI scans combines both imaging modalities, simultaneously providing structural and functional information in assessing tumors. Incorporating PET with MRI techniques provides the structural and anatomical data from MRIs, with additional metabolic and morphological information from PET scans. Although hybrid PET/MRI scans have immense usefulness in diagnosing brain tumors, limitations include longer processing times, less availability, and higher costs [123]. For these reasons, PET/CT scans are more common in today’s practice. However, in cases where it may be challenging to discern between grades of gliomas and differentiate between treatment-related changes, PET/MRI hybrid scans may be helpful.

Integrating an AI system enhances the diagnostic potential of PET/MRI imaging. Using a convolutional neural network with PET, MRI, and combined PET/MRI imaging, hybrid PET/MRI achieved a recognition accuracy of 97% in determining the extent of gliomas [124]. Despite this high accuracy, the hybrid modality demonstrated low specificity, leading to a high misdiagnosis rate. Future advancements must address reducing misdiagnosis while maintaining high accuracy, further improving the diagnostic utility of this combined modality. Advanced imaging techniques clarify tumor characteristics, enabling the identification of unique morphological features, refining diagnostics, and guiding treatment strategies.

## 5. Biomarker Integration in Neuro-Oncology: A Brief Background

While imaging allows for the physical detection of tumors and monitoring during treatment, biomarkers are equally crucial in tumor evaluation. Certain biomarkers are essential in diagnosing specific brain tumors and provide valuable information regarding a tumor’s pathology and biochemical pathway. Furthermore, it is challenging to classify tumors based on imaging or histology alone, so biomarkers help to confirm tumor diagnosis. Integrating laboratory markers as a diagnostic tool has proven significant in determining patient outcomes by allowing for more accurate genetic and molecular profiling of tumors. The most extensively studied critical biomarkers in brain tumors include 1p/19q co-deletion, IDH isozyme mutation, MGMT methylation status, and *TERT* promoter mutation [125,126,127]. In this section, we describe the current biomarkers in detecting brain tumors and the potential of using liquid biopsies as a novel diagnostic tool. A summary of the listed biomarkers is presented in Table 3.

### 5.1. 1p/19q Co-Deletion

A 1p/19q co-deletion, the loss of the short arm of chromosome 1 and the long arm of chromosome 19, has predicted a better prognosis and better survivability for oligodendroglioma patients. The oligodendroglioma (WHO grade II), a diffuse infiltrating low-grade brain tumor that develops in the white matter of the cerebral hemisphere, utilizes biomarkers to predict patient outcomes [128]. Standard 1p/19q co-deletion detection involves fluorescent in situ hybridization (FISH) and polymerase chain reaction (PCR) [129]. Identifying key biomarkers, such as the 1p/19q co-deletion, has paved the way for better-targeted therapies and aided in early recognition and treatment of such tumors. Tumors with the 1p/19q co-deletion have a better prognosis compared to tumors that do not have the co-deletion. Patients with the 1p/19q co-deletion had more remarkable survival, making it an essential prognostic factor [130]. Additionally, there is evidence that glioma tumor contour can reveal the presence of a 1p/19q co-deletion [131] These advancements in neuro-oncology emphasize the importance of tools that enhance precision and efficacy in detecting this co-deletion.

### 5.2. Isocitrate Dehydrogenase (IDH) Isozyme Mutation

The isocitrate dehydrogenase (IDH) isozyme mutation is a critical biomarker for diagnosing brain tumors. The IDH isozyme is one of the most significant prognostic factors for overall survival in diffuse gliomas [132,133]. Patients with *IDH* mutation typically have better outcomes than patients with IDH1/2 wildtype variants [134,135]. Current *IDH* mutation detection involves direct sequencing, immunohistochemistry, MRS, and monoclonal antibodies [136,137,138]. The *IDH* isozyme mutation illustrates the increasing trend toward targeting molecular biomarkers as a diagnostic and prognostic tool for CNS tumors, which can be detected through liquid biopsy.

### 5.3. O^6^-Methylguanine-DNA Methyltransferase (MGMT) Methylation Status

The O^6^-methylguanine-DNA methyltransferase (MGMT) methylation status is another important clinical biomarker and prognostic factor for GBM. Specifically, the MGMT methylation status is a prognostic factor indicating a better response to TMZ and radiation therapy, leading to higher overall survival in patients [139,140]. The conventional detection of MGMT methylation status is achieved using several methods, such as non-quantitative methylation-specific polymerase chain reaction (MSP), quantitative methylation-specific PCR (MSP), immunohistochemistry, and pyrosequencing [141]. These methods may not reliably monitor GBM patients. Techniques such as methylation-specific PCR and pyrosequencing predict survival following TMZ treatment more effectively than immunohistochemistry [142]. Other diagnostic approaches, such as liquid biopsies, could further enhance the detection of MGMT in GBM diagnosis [142].

### 5.4. Telomerase Reverse Transcriptase (TERT) Mutation

Telomerase reverse transcriptase (TERT) is a catalytic subunit of telomerase in normal tissue that is typically silenced to prevent oncogenesis [143]. Mutations in the *TERT* promoter increase tumor aggressiveness due to continuous activation [143]. Detection of *TERT* promoter mutations conventionally involves Sanger sequencing and next-generation sequencing (NGS) [144]. NGS is far less expensive and more efficient than whole genome sequencing because it utilizes a broader genomic region. However, NGS can present inaccuracies due to the fast turnaround time and broad genome analysis [145]. Therefore, while helpful, it is essential to interpret results with caution. The *TERT* mutation is a critical biomarker for brain tumors, and conventional methods may prove beneficial to some extent, with some limitations being that they are more expensive and take more time.

Including key biomarkers such as enzyme mutations, methylation status, and promoter mutations has enhanced the personalization and effectiveness of GBM treatment. These biomarkers are beginning to be clinically used to predict overall survival in patients with GBM, and many diagnostic tools aim to make diagnosis accurate and early. Identifying these key biomarkers enables clinicians to develop treatment plans tailored to specific tumor types. Advances in laboratory techniques have improved biomarker detection, with liquid biopsies playing a primary role.

## 6. Liquid Biopsy: A Non-Invasive Frontier in Neuro-Oncology Diagnostics

Imaging remains the primary non-invasive method for initial tumor evaluation; recent advances have highlighted biomarkers as valuable diagnostic tools. Although imaging identifies tumors, more complex cases may warrant further concrete diagnostic tools, such as tissue biopsy and laboratory techniques, for additional investigation. A significant challenge is that the classification of a tumor may require invasive techniques, such as sampling a biopsy from the brain or spinal cord, to identify histological features. Although the literature suggests that brain biopsy is relatively safe, potential iatrogenic complications persist [146]. Currently, liquid biopsies are not included in standard neuro-oncological evaluations, but they hold significant promise in becoming a powerful diagnostic tool. Even more so, liquid biopsy may be utilized in complex occasions where radiation necrosis and pseudoprogression can be commonly confused with true tumor progression [147]. Less invasive techniques, such as liquid biopsy, analyze these biomarkers, offering a potentially more versatile approach to managing and monitoring brain tumors.

Liquid biopsy originated with identifying free circulating tumor DNA (ctDNA) in the serum of cancer patients [148]. Tumors secrete components, including ctDNA, into the bloodstream, allowing for quantification. Since its discovery, research has focused on exploring the clinical applications of ctDNA and its relationship to tumor biology. Liquid biopsy does not require invasive techniques, such as sampling brain tissue, focusing on using biomarkers to identify tumors early. The process of a liquid biopsy involves collecting a sample of body fluid, serum, or cerebrospinal fluid (CSF) and analyzing this fluid, as depicted in Figure 2. Different substances, such as ctDNA, extracellular vesicles (EVs), circulating tumor cells (CTCs), and tumor-educated platelets (TEPs), are extracted and analyzed to provide key insights into diagnosis and management of CNS tumors [149].

This figure emphasizes the multifaceted capabilities of liquid biopsy in advancing neuro-oncology diagnostics. It offers a non-invasive alternative to traditional biopsy techniques and showcases the integration of molecular and cellular data to improve patient care and treatment outcomes.

Additionally, liquid biopsy, in conjunction with imaging, is a cost-effective and efficient diagnostic tool [149]. Conventional tissue biopsy is more sensitive and provides definitive information about a tumor, but notable disadvantages exist. Brain tissue biopsy requires more time, expertise, and resources, and it carries a risk of dangerous complications like tumor seeding. On the other hand, liquid biopsy is generally non-invasive, less expensive, and lowers the risk of tumor seeding [149]. These characteristics make it a convenient alternative to tissue biopsy in certain circumstances where brain biopsy may be dangerous, but there are limitations with liquid biopsy. One limitation is that early-stage tumors may secrete low, undetectable concentrations into the blood or CSF or may not even shed components [150]. However, liquid biopsy is still a viable option in brain tumor diagnostics because it may be quicker and more accessible compared to tissue biopsy.

### 6.1. Free Circulating Tumor DNA (ctDNA): A Window into Tumor Dynamics

Early cancers release free ctDNA into the serum, with brain tumors typically exhibiting higher levels due to their high cell turnover. One key application of ctDNA is the identification of primary GBMs [151]. Detectable levels of ctDNA were found in nearly half of primary brain tumor patients, with 48.9% exhibiting targetable therapy options in the genome. ctDNA shows significant potential as a non-invasive biomarker for tumor detection and developing personalized treatment plans. It also monitors tumor progression and treatment response [152,153]. Unlike tissue biopsy, which provides only a snapshot of a tumor’s condition, ctDNA enables continuous monitoring of brain tumors over time.

Liquid biopsy has proven effective in distinguishing actual progression from pseudoprogression in diffuse midline glioma patients [154]. In a completed clinical trial, patients with diffuse midline glioma harboring the H3 K27M mutation received the investigational drug imipiridone ONC201 (NCT03416530) [155]. Liquid biopsy effectively monitored treatment response through the analysis of ctDNA in CSF and plasma samples. This trial is notable as no prior studies have demonstrated the efficacy of ctDNA in predicting treatment response or pseudoprogression. Results showed that some patients exhibited a spike in ctDNA levels preceding tumor progression [154]. However, the sample size was limited to 24 children. Incorporating ctDNA as a diagnostic tool may predict tumor changes, enabling providers to plan treatments and prevent complications.

Liquid biopsy techniques have analyzed the methylation profile of ctDNA in pediatric patients to diagnose brain tumors [156]. Seven of 20 samples were correctly classified, with tumors exhibiting higher ctDNA fractions and burdens consistently identified. Low ctDNA levels often led to misclassification. These results demonstrate the potential of ctDNA as an alternative diagnostic tool. Increasing the sample size and including more diverse populations could validate ctDNA for brain tumor diagnostics [156].

### 6.2. Unveiling Tumor Biology Through Extracellular Vesicles (EVs)

Extracellular vesicles are another measured component of liquid biopsies that offer a wealth of information about brain tumors. Tumor cells release EVs, small membrane particles, into the extracellular space. These particles play a key role in cell communication and provide insight into the cancer state [157]. More importantly, EVs can cross the BBB, containing information about metabolites and macromolecules such as proteins, lipids, nucleic acids, and sugars. Specifically, the mRNA inside EVs codes for important protein biomarkers such as IDH enzyme, p53, and growth factor receptors [158]. These biomarkers obtained from EVs help identify brain tumor diagnosis, aiding in early detection, monitoring, and treatment strategies. Several studies are developing serum panels to diagnose GBM.

An EV-based panel distinguishes *IDH* isozyme mutations, *MGMT* promoter methylation, *TERT* promoter mutation, and p53 mutations in GBM [159]. Sequencing and quantifying RNA isolated from GBM EVs revealed significant differences compared to EVs in healthy patients. The EV-based panel demonstrated high sensitivity and specificity for key biomarkers: *IDH1* (96% sensitivity, 80% specificity), *MGMT* methylation (91% sensitivity, 73% specificity), p53 (100% sensitivity, 89% specificity), and *TERT* promoter mutation (89% sensitivity, 100% specificity). These biomarkers provide critical insights into disease prognosis.

A new staining method with Pyronin Y detects DNA associated with EVs and ctDNA in the plasma [160]. DNA profiles in GBM patients were hypothesized to differ from those in healthy patients. Nucleic acid levels in GBM patients were increased in EVs and plasma samples; however, they were only significantly increased in the plasma samples [160]. Furthermore, measuring extracellular vesicles indicates how well a patient responds to therapy and reveals changes associated with response to treatment [161,162]. Future work should explore EV’s potential in diagnostics and validate its results.

### 6.3. Circulating Tumor Cells (CTCs): Tracking Tumor Progression and Metastasis

Circulating tumor cells (CTCs) serve as critical biomarkers, providing insights into the metastatic progression of brain tumors. CTCs detach from primary tumors, enter the bloodstream, and drive metastasis [163,164]. Detecting CTCs facilitates early diagnosis, preventing tumor progression and improving overall patient survival. Serum analysis has already demonstrated the utility of CTC detection in bladder carcinoma [165]. A telomerase-based CTC assay evaluates telomerase activity to diagnose high-grade brain tumors [165]. Spiral microfluidic technology isolates CTCs from whole blood in GBM patients before and after surgery, providing a real-time method to monitor tumor status [166]. Elevated CTC levels indicate greater severity of GBM, with higher levels correlating directly with increased malignancy [167]. CTC detection offers a non-invasive approach to characterizing brain tumors, aiding prognosis, and optimizing patient outcomes.

### 6.4. Tumor-Educated Platelets (TEPs) as Diagnostic Biomarkers

TEPs are platelets in the microenvironment of cancer cells influencing tumorigenesis through direct and indirect interactions [168]. Exposure to cancer cells alters platelet behavior, leading to changes in cancers containing altered mRNA expression profiles that provide information about brain cancers [169]. Through direct interactions, adhesion molecules such as glycoproteins on platelets and tumor cells may be upregulated, influencing tumor growth. Platelet interaction through specific adhesion molecules, including glycoproteins, selectins, and integrins, is involved in “educating” platelets [169]. Additionally, P-selectin, a valuable marker on the platelet membrane, also interacts with tumor cells, promoting tumorigenesis [170]. Through indirect interactions, the release of signaling molecules contributes to tumorigenesis. Platelets in GBM patients showed higher vascular endothelial growth factor receptor (VEGFR-1) expression, promoting pro-angiogenic effects [171]. Therefore, several novel diagnostic tools are aimed at utilizing TEPs as a biomarker for brain tumors. A multicenter study developed a TEP RNA panel to detect and monitor GBM patients, demonstrating that this blood test enhances differentiation between pseudoprogression and true progression with an accuracy of 85% [172].

### 6.5. AI-Powered Insights: Advancing Liquid Biopsy for CNS Tumors

Supplementing liquid biopsy with AI can improve workflow and enhance diagnostic accuracy. Several benefits of using AI include early detection, precise diagnosis, and faster results [173,174]. Many ML programs exist, but a common theme is utilizing complex algorithms to analyze liquid biopsy data and predict tumor detection with high accuracy [174,175]. Not many studies exist evaluating AI and liquid biopsies in brain tumors. Tumors can be detected from urine using an AI-based approach [176]. Research regarding AI and liquid biopsy should be directed toward developing a standard method for diagnosing tumors, as few studies describe this. Including AI in the analysis of liquid biopsies may be a more advanced novel technique that aids in detecting brain cancers and interpreting complicated results.

### 6.6. Considerations for Liquid Biopsy

To allow for the significant potential of liquid biopsy in the diagnosis and evaluation of brain tumors to be appropriately utilized, several limitations of the technology must be addressed. Among these is the BBB, which limits communication between the brain and blood, physiologically limiting the presence of cells in the blood. If tumor tissue does not sufficiently disrupt this barrier, biomarkers may not be found within liquid biopsy samples [177]. One potential workaround for this brain-specific challenge is the utilization of focused ultrasound to temporarily open the BBB, allowing for leakage of biomarkers into the blood that can be sampled for liquid biopsy. However, this remains an important area of research [178].

While initial implementation of this technology and training may incur costs, it appears that in many cases involving various cancer types, liquid biopsy is generally cost-effective in managing and selecting treatment for multiple tumors [179]. However, this remains to be investigated for liquid biopsy as a prognostic tool or in evaluating the risk of relapse [179]. Finally, another challenge of liquid biopsy is standardizing the techniques used. Although liquid biopsy has generally been shown to be able to detect tumors early, it is essential to use this technology in an appropriate time frame, as some biomarkers may not be at sufficient levels within the blood to be detected very early in the disease course, potentially leading to false negatives [180]. Selecting patients who are appropriate for assessment with liquid biopsy will also heighten its utility as a diagnostic tool. Additionally, appropriate biomarkers need to be implemented to mitigate potential issues with sensitivity and specificity from liquid biopsy [180]. As such, these challenges require increased research into liquid biopsy to develop guidelines about when and how often liquid biopsy should be used in the diagnostic timeline, which patients will benefit from it, and which biomarkers are the most appropriate.

## 7. Artificial Intelligence in Neuro-Oncology: Enhancing Diagnostics and Precision Medicine

With the advancements seen in advanced MRI modalities and radiotracers, classifying brain tumors using imaging still presents challenges, especially with interobserver bias. Moreover, making an accurate diagnosis is difficult with more complex cases, outdated machines, and varying expertise levels of clinicians [181]. Diagnostic modalities and AI have grown substantially in recent years, making accurate diagnoses and earlier detection feasible. Including AI in diagnostic imaging and tumor molecular profile analysis is a growing research field. Its primary use in practice is to reduce interobserver bias and measurement errors, increase the speed of analysis, and detect tumors with higher accuracy [182]. AI has been applied increasingly across different medical specialties, including neuro-oncology, improving healthcare diagnostics. Leveraging the data provided by advanced imaging and liquid biopsy in combination with the processing power of AI will help reliably and effectively use the wealth of data provided by these techniques. Here, we will discuss the potential of different forms of AI in its use with the combination of diagnostic techniques discussed earlier.

### 7.1. Machine Learning (ML), Deep Learning (DL), and Convolutional Neural Networks (CNNs): Revolutionizing Tumor Diagnostics

The evolution of AI programs has led to the development of different subsets: machine learning (ML), deep learning (DL), convolutional neural networks (CNNs), and transformer-based AI (TBAI). While these terms are related, they differ in their processes and applications. AI refers to computer systems that simulate human intelligence, including problem-solving, decision-making, and pattern recognition skills [183]. ML is a subset of AI, utilizing algorithms to identify patterns and build models from existing data [183]. The program then uses this data to predict a condition or disease, providing a more informed conclusion about tumor diagnosis, morphology, and therapy response [184]. DL is a further subset of ML, utilizing specialized algorithms capable of more complex decision-making processes with greater autonomy and requiring less human intervention [185,186,187,188]. CNN is a further refinement of DL, processing large volumes of image-based data more efficiently and autonomously [189]. Meanwhile, TBAI is another form of DL that focuses on using an “attention” mechanism, where it can look at a wide variety of datasets and draw attention to the essential parts of each to come to an overall decision [190,191].

Figure 3 visually represents this hierarchical framework and application of AI, illustrating the relationships among its subsets and their specific applications in neuro-oncology. It highlights TL’s pivotal role in enhancing these AI technologies by adapting pre-trained models. This diagram demonstrates how the AI framework improves diagnostic capability by automating tumor detection, improving accuracy, and streamlining workflow. This integration is critical in supplementing clinical decision-making, enabling more precise diagnoses.

### 7.2. Reducing Diagnostic Errors: AI in Imaging and Tumor Detection

Supplementation of an AI system with provider judgment has improved diagnostic accuracy in other medical specialties. One example of AI’s capacity for improving cancer diagnosis is detecting breast cancers through an AI-based decision system in ultrasound [192]. In this analysis, three American Board of Radiology-certified radiologists reviewed ultrasound images without an AI-based decision system and scored the case via Breast Imaging Reports and Data System and the likelihood of malignancy. This served as their control. After a 4-week washout, they evaluated another set of ultrasound cases with AI-based decision system outputs. The AI system was presented in two ways: sequentially, where it aided after the unassisted reading, and independently, where both the case and AI output were reviewed together. While sequential presentation showed no significant difference, the independent strategy led to a statistically significant improvement in diagnostic accuracy. For example, one radiologist’s AUC score improved from 0.7618 without AI assistance to 0.8213 using the independent AI system [192]. Adding AI as a diagnostic tool could only enhance accuracy and reduce variability. AI, which has demonstrated significant potential in breast cancer detection, could improve tumor detection and the prediction of treatment-related changes in neuro-oncology [193].

Although different AI programs exist, they detect tumors more efficiently, especially in complex cases. Specifically, incorporating AI could be helpful when it is difficult to differentiate between primary and metastatic tumors. A DL algorithm, SCAT (Spatial Convolutional Attention Inception), was used to distinguish between a single metastasis and high-grade GBM, which are commonly confused [194]. Using a DL algorithm to detect these malignancies, clinicians reduced errors and improved accuracy to 92.3% [194]. Several studies utilizing DL with different diagnostic modalities in differentiating GBM from metastatic lesions showed improvement in the same effect [195,196].

CNNs supplement imaging diagnostics to detect different glioma grades with minimal human input, potentially improving workflow and efficacy [197]. A DL algorithm applied to 617 pediatric participants accurately detected posterior fossa brain tumors [198]. These results underscore the effectiveness of DL and CNNs in distinguishing challenging brain tumor types, such as metastases versus high-grade GBM, and differentiating glioma grades.

### 7.3. Transfer Learning: Addressing Rare Tumor Challenges

Despite these advantages, AI’s status as a new technology is a drawback. While the primary purpose of AI is to enhance accuracy in brain tumor detection, there remains a scarcity of studies demonstrating its efficacy in predicting tumors. These programs rely on extensive data and training for accurate predictions, with professionals frequently cross-checking the results. An issue is that this data may not be available, especially for rare brain tumors. One way to combat this challenge is through transfer learning (TL), a pre-trained program used to process medical images. TL is an ML technique where data obtained from one problem is transferred and applied to solve a different task [185]. TL is used in the subdivisions of recurrent neural networks, general adversarial networks, and CNNs [199]. However, TL has mainly been described in CNNs, as this subdivision has been more extensively studied. While the clinical use of TL has rarely been described in neuro-oncology, there have been few studies on the use of TL in other fields of medicine, such as pulmonology [200]. With the growing use of AI in neuro-oncology, TL utilizes pre-trained models to reduce extensive training while still providing high-quality decision-making support [201]. Ultimately, AI’s application in neuro-oncology is an avenue for research that could serve as a purposeful tool in diagnostics.

### 7.4. Extracting Tumor Features with Radiomics: Towards Personalized Medicine

A future direction for AI as a diagnostic technique is through radiomics. Radiomics is defined as the extraction of quantitative features from radiological images used to predict clinical outcomes [202,203]. The process is complex and typically requires multiple processing steps. Typically, data are extracted from diagnostic images such as MRI, CT, and PET. The radiomic features, such as tumor characteristics, are extracted and analyzed through computer programming and ML. The data gathered are then compiled and combined with patient data such as demographic, genomic, and histologic data [203]. It is then analyzed using algorithms, creating a more personalized approach to therapy [204]. Radiomics is especially useful in detecting characteristics in complex cases because it uncovers details that may not have been noted otherwise. These characteristics include shape and heterogeneity [203]. A critical aspect of radiomics is that it aids in assessing treatment response and detecting brain tumors [205,206].

### 7.5. Transformer-Based AI: Combining Data to Create a Full Picture

It is essential to highlight different datasets to create a complete clinical picture for a patient’s diagnosis and evaluation. Transformer-based AI (TBAI) systems can incorporate different information to assist in this evaluation. TBAI utilizes a unique “attention” mechanism to determine the most critical data. Often, it is trained on vast datasets and then can incorporate transfer learning to be fitted to specific tasks [190]. The capacity of TBAI has been shown to evaluate different tumors. For example, ROAM (Region-based Attention with Multi-scale Learning) is a transformer-based system that showed efficacy in tumor detection, grading, subtyping, and predicting molecular markers like IDH mutation and MGMT promoter methylation based on histopathology slides [207]. Another TBAI showed high precision and accuracy in evaluating and diagnosing brain tumors based on internal and publicly available MRI datasets [208]. A third study on public datasets across three institutions produced a glioblastoma survival model that incorporated multimodal MRI, clinical, and molecular-pathologic data with similar findings across the datasets [209]. These studies highlight the potential functionality of TBAI to enhance and employ existing diagnostic modalities to predict and provide information that can alter the evaluation of tumors based on various data. With the development of other diagnostic modalities, their incorporation into TBAI can potentially impact the landscape of brain tumor diagnostics.

### 7.6. Ethical and Practical Barriers to AI

While the idea of AI in clinical practice is promising, some ethical considerations and barriers remain in question. Ethically, there are significant concerns about maintaining patient privacy and data security, as AI systems require large datasets involving sensitive patient information. Faulty security systems, cyberattacks, or data misuse can foster distrust in medical AI systems. Suggestions to ensure patient confidentiality when using AI may include multiple levels of encryption, strict oversight when sharing data, and privacy-preserving systems [210,211,212]. Over-reliance on AI systems could affect diagnosis and undermine patient safety. Providing training and continuous education for healthcare providers on effectively using AI models as a tool rather than a replacement for decision-making may avoid over-reliance concerns. When mistakes occur, transparency through precise documentation and records determines liability, which may involve multiple entities such as developers, providers, and institutions [213].

Practical barriers to AI include availability, cost, and quality of data. One primary hurdle in integrating AI into neuro-oncology is ensuring these systems are compatible with existing healthcare tools. Implementing AI models will incur extra costs, especially if machines are noncompatible or may require software updates. Additionally, additional costs may include required training for staff and licensing, which is challenging in places where funding may not be readily available. The financial burden associated with AI may be reduced through government incentives or reimbursement from insurance companies [214]. Another challenge with AI is standardizing the vast number of existing models. AI systems must be trained on extensive and diverse patient populations to be effective. However, it may be difficult to achieve quality data, particularly from underrepresented groups or rare cases. Furthermore, deciding which specific data would be included when training an AI system may be difficult. The varying methods and lack of uniform standards for integrating AI systems contribute to the poor quality of data [215]. Despite these challenges, data quality can be improved by combining information from diverse sources and taking precautions to ensure the data are accurate and representative.

## 8. Shaping the Future of Neuro-Oncology Diagnostics: Challenges and Opportunities

Neuro-oncology has seen substantial advances in the past decade, with cutting-edge diagnostic tools becoming integral to early detection, characterization, and differentiation of pseudoprogression and radiation necrosis from actual progression. Conventional imaging methods, such as traditional MRI, remain essential for tumor diagnosis but now work alongside more sophisticated modalities, including DTI, MRS, PET radiotracers, hybrid PET/MRI scans, and AI. Additionally, integrating biomarkers with imaging techniques has further enabled more precise diagnosis. Expressly, liquid biopsies and analysis of their components, ctDNA, EVs, CTCs, and TEPs, have provided unique insights into tumorigenesis of CNS tumors, further refining patient management.

These advances in diagnostic tools enable more precise diagnoses. Earlier detection and more accurate diagnosis of brain tumors personalize treatment strategies, leading to better prognoses and faster treatment for patients. Future research on diagnostic neuro-oncological tools should focus on integrating these modalities into clinical practice and validating their use in interpreting complex brain tumor cases. Research should investigate the clinical utility of these tools in distinguishing tumor progression, radiation necrosis, and pseudoprogression, which are common challenges in the field.

However, despite the advancement of such tools, challenges persist. The availability and accessibility of these novel tools remain scarce. Higher-quality imaging scans, such as advanced MRI techniques and PET/MRI hybrid scans, typically cost more and are not as readily available as conventional MRI. Additionally, because these novel diagnostic tools are relatively new, research must focus on the reliability and validity of these diagnostic techniques.

With the development of liquid biopsy, panels have no standardization in detecting and interpreting EVs, CTCs, TEPs, and ctDNA. This calls for more research regarding the validity of panels surrounding liquid biopsies. Specific diagnostic tools, such as the radiotracers,^18^F-FPIA, ^18^F-FET, and ^18^F-FACBC, are undergoing clinical trials exploring their efficacy in tumor diagnosis. Additionally, further research should reduce inter-observer bias through AI and improve diagnostic efficacy using AI subsets such as ML, DL, and CNN. While much more research is needed, the diagnostic tools in neuro-oncology are promising.

These advances in imaging, AI, and liquid biopsies represent a change in transforming patient care, transitioning to more non-invasive methods and early detection. Identifying CNS tumors with high precision aids in more accurate diagnosis and personalized treatment, improving patient outcomes.

## 9. Conclusions

Advances in imaging technologies, liquid biopsy techniques, and AI are transforming neuro-oncology diagnostics. These tools tackle challenges such as tumor detection, characterization, and the differentiation of treatment effects, like pseudoprogression and radiation necrosis. Clinicians now use advanced imaging techniques, including diffusion tensor imaging (DTI), magnetic resonance spectroscopy (MRS), and hybrid PET/MRI, to identify tumor structures, analyze metabolism, and track progression. Radiotracers such as ^18^F-FPIA, ^18^F-FET, and ^18^F-FACBC improve diagnostic accuracy, especially in complex cases.

Liquid biopsy tools, including circulating tumor DNA (ctDNA), extracellular vesicles (EVs), circulating tumor cells (CTCs), and tumor-educated platelets (TEPs), provide real-time, non-invasive ways to monitor tumors. These biomarkers support early detection, guide treatment decisions, and enable personalized strategies. AI-driven analytics reduce diagnostic errors, speed up workflows, and sharpen the precision of clinical decision-making.

Clinicians and researchers must address challenges such as accessibility and integrate these tools into clinical practice. They must also focus on validating technologies, standardizing their use, and solving equity gaps to ensure widespread benefits. Prioritizing collaboration between technology, research, and practice enables neuro-oncology teams to deliver earlier diagnoses, precise treatments, and improved outcomes for patients with central nervous system tumors.

## Figures and Tables

**Figure 1 ijms-26-00917-f001:**
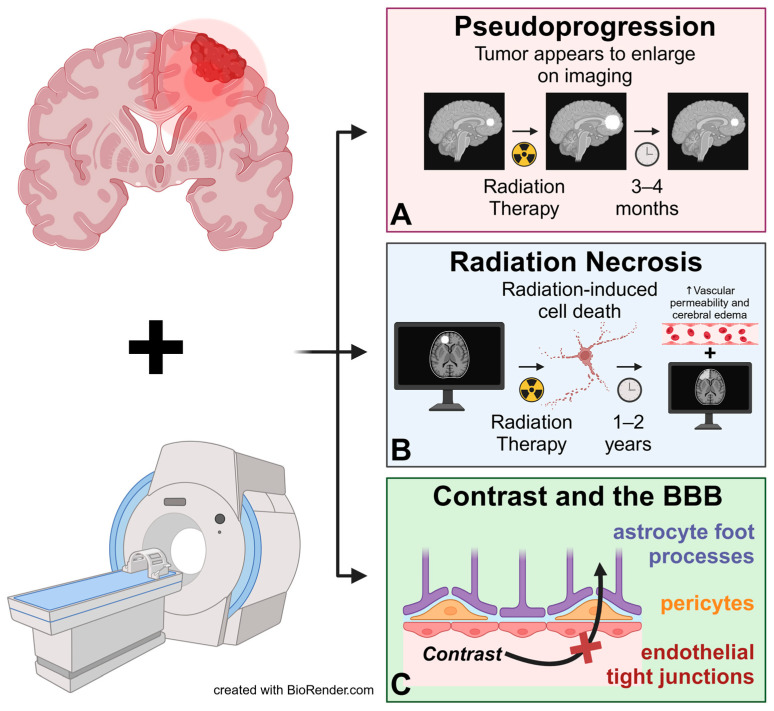
Post-therapy MRI challenges in differentiating tumor progression and treatment effects. The main obstacles in assessing brain tumors using MRI post-treatment are presented as follows: (**A**) Pseudoprogression: Post-radiation inflammation can cause temporary tumor enlargement on imaging, mimicking actual tumor progression. Radiation therapy triggers increased vascular permeability and inflammatory responses, causing this phenomenon, which typically resolves within 3–4 months. Misinterpretation as true progression can complicate diagnosis and delay appropriate management. (**B**) Radiation Necrosis: A delayed complication of radiation therapy, radiation necrosis manifests as cell death, increased vascular permeability, and cerebral edema, usually appearing 1–2 years post-treatment. These pathological changes can create imaging findings that resemble tumor recurrence, complicating diagnosis and treatment planning. Differentiating radiation necrosis from true tumor progression often requires advanced imaging techniques or complementary diagnostics. (**C**) Contrast and the Blood-Brain Barrier (BBB): The BBB, consisting of endothelial tight junctions, pericytes, and astrocytic foot processes, regulates the passage of substances into the brain. Common MRI contrast agents like gadolinium may struggle to penetrate the intact BBB, particularly in deep-seated or partially infiltrative tumors. This limitation reduces imaging sensitivity and may obscure tumor boundaries, necessitating alternative or adjunctive diagnostic methods.

**Figure 2 ijms-26-00917-f002:**
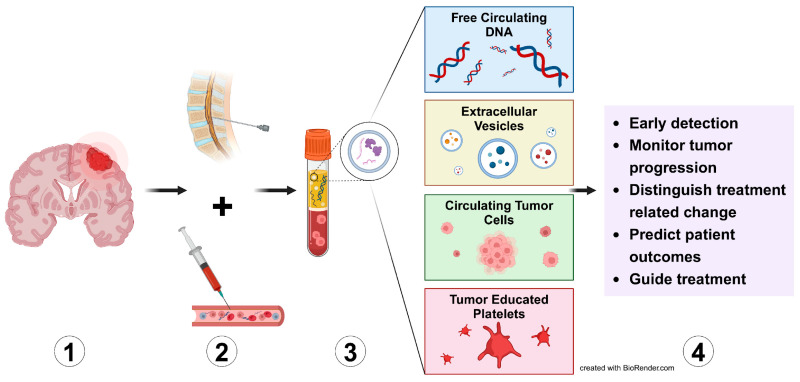
Overview of liquid biopsy components and applications in neuro-oncology. The process and key components of liquid biopsy, a minimally invasive diagnostic tool for neuro-oncology, are illustrated, highlighting its potential for early detection and patient management. (**1**) Tumor Origin: Tumors within the central nervous system (CNS) shed molecular and cellular components into surrounding biological fluids, such as blood and cerebrospinal fluid (CSF). These materials are subsequently released into circulation. (**2**) Sample Collection: Liquid biopsy involves the collection of patient blood or CSF through minimally invasive techniques. This allows for the isolation of tumor-derived biomarkers without requiring direct tissue sampling. (**3**) Key Biomarkers: Free Circulating DNA (cfDNA): Tumor cells release DNA fragments into the bloodstream, carrying tumor-specific mutations or methylation patterns. Extracellular Vesicles (EVs): Tumor cells release nanoparticles containing proteins, RNA, and DNA that reflect tumor biology and progression. Circulating Tumor Cells (CTCs): Intact tumor cells that escape from the primary tumor into circulation, providing direct cellular information about the cancer. Tumor-Educated Platelets (TEPs): Tumor cells reprogram platelets to reflect tumor-specific RNA and protein signatures, aiding in tumor detection and characterization. (**4**) Clinical Applications: The liquid biopsy components have a wide range of diagnostic and prognostic uses, including: Early Detection: Identifying tumors before clinical symptoms arise. Monitoring Tumor Progression: Tracking changes in tumor biology over time. Distinguishing Treatment-Related Changes: Differentiating true tumor progression from pseudoprogression or radiation necrosis. Predicting Patient Outcomes: Assessing tumor aggressiveness and potential treatment responses. Guiding Treatment: Informing personalized therapeutic strategies based on real-time tumor monitoring.

**Figure 3 ijms-26-00917-f003:**
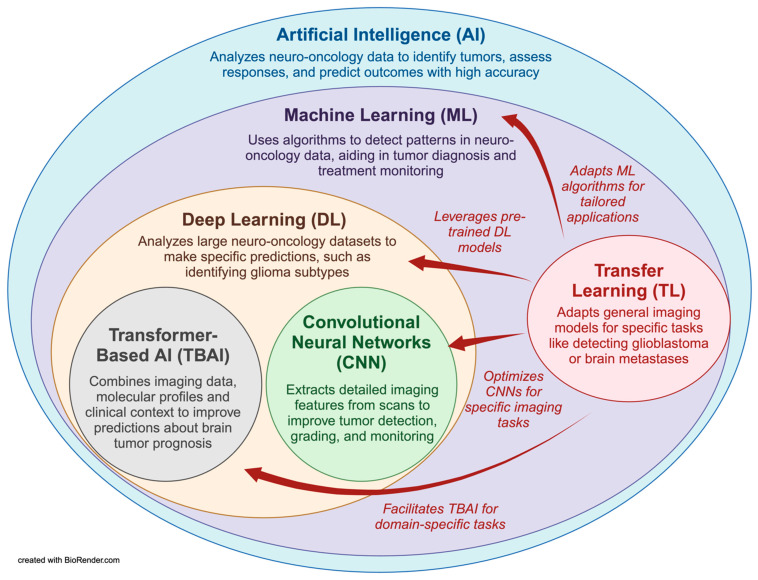
Hierarchical framework and applications of artificial intelligence in neuro-oncology diagnostics. The hierarchical relationship between artificial intelligence (AI), machine learning (ML), deep learning (DL), convolutional neural networks (CNNs), transformer-based AI (TBAI), and transfer learning (TL) is represented, focusing on their applications in neuro-oncology diagnostics. AI forms the broadest layer, encompassing techniques for analyzing complex neuro-oncology datasets to identify tumors, assess treatment responses, and predict patient outcomes. ML represents a subset of AI that uses algorithms to detect patterns in data, such as MRI scans or biomarker profiles, to aid in tumor diagnosis and treatment monitoring. DL, a specialized subset of ML, leverages large neuro-oncology datasets to make specific predictions, such as identifying glioma subtypes. CNNs, a distinct architecture within DL, extract detailed imaging features from scans (e.g., MRI, PET) to improve tumor detection, grading, and monitoring. TBAI is another subset of DL that utilizes an attention mechanism to identify the most crucial information, excelling in the context of large datasets and a variety of data to improve predictions about prognosis and tumor progression. TL is depicted as a technique that intersects these frameworks, adapting pre-trained models for specific neuro-oncology tasks, such as distinguishing glioblastoma from brain metastases. The arrows originating from TL represent its role in optimizing the performance of ML, DL, CNN, and TBAI: To ML: TL adapts existing ML algorithms for tailored diagnostic applications. To DL: TL leverages pre-trained DL models to enhance predictions with limited data. To CNNs: TL optimizes CNN architectures for specialized imaging tasks, improving accuracy and efficiency. To TBAI: TL allows TBAI to be focused on domain-specific tasks based on available data types.

**Table 1 ijms-26-00917-t001:** Description of radiotracers used in PET Imaging.

PET Radiotracer	^18^F-Fluorodeoxyglucose (^18^F-FDG)	^18^F-Fluoropivalate (^18^F-FPIA)	^18^F-Fluoroethyltyrosine (^18^F-FET)	^18^F-Fluciclovine (^18^F-FACBC)
Structural Analog	Glucose	Short Chain Fatty Acid	Amino Acid	Amino Acid
Proposed Use	Identify tumors with high glucose metabolism	Target fatty acid metabolism in high-grade gliomas	Exploit increased amino acid uptake via LAT transporters	Exploit increased amino acid uptake via LAT transporters
Clinical Significance	- Most used radiotracer for glycolysis detection - Widely researched	- Identify high-grade gliomas - Differentiate tumor grades - Monitor treatment effects	- Differentiate true progression from treatment-related changes	- Enhanced tumor-to-background contrast - Distinguish progression from radiation necrosis

This table describes the structural characteristics, the proposed use, and the clinical significance of radiotracers being studied in brain tumors.

**Table 2 ijms-26-00917-t002:** Current clinical trials for radiotracers.

PET Radiotracer	Trial Code	Trial Title	Aims/Outcomes
^18^F-FPIA	NCT05801159 [104]	[^18^F]FPIA PET-CT in Glioblastoma Multiforme (GBM) (FAM-GBM)	- Using ^18^F-FPIA to assess the degree of fatty acid metabolism in GBM patients after treatment with chemoradiotherapy - Detecting treatment-related changes - Measuring ^18^F-FPIA uptake in tumors
^18^F-FET	NCT01756352 [112]	FET-PET for Evaluation of Response of Recurrent GBM to Avastin	- Predicting progression-free survival using FET-PET in recurrent GBM patients receiving Avastin
^18^F-FACBC	NCT05554302 [117]	Characterization of ^18^F-Fluciclovine PET Amino Acid Radiotracer in Resected Brain Metastasis (CONCORDANT)	- Assessing the extent of surgery in brain metastasis patients using ^18^F-Fluciclovine - Determining if ^18^F-Fluciclovine can detect residual tumors post-operatively beyond what MRI alone can identify - Earlier detection of recurrent tumors using ^18^F-Fluciclovine
	NCT06159335 [103]	^18^F-FLUC-CEST PET/MR in Patients With Brain Mets	- Measuring ^18^F-Fluciclovine uptake in brain tumors with PET - Measuring cytosolic proteins in brain tumors using MR CEST
	NCT04462419 [118]	^18^F-fluciclovine PET/MRI Imaging for the Detection of Tumor Recurrence After Radiation Injury to the Brain	- Differentiating true tumor progression from radionecrosis in metastatic brain tumor patients who were previously treated with radiation therapy

This table summarizes current clinical trials studying various radiotracers and their outcomes.

**Table 3 ijms-26-00917-t003:** Common biomarkers for brain tumors.

Biomarker	Method of Detection	Clinical Significance
1p/19q Co-deletion	FISH, PCR	Associated with better prognosis and survival in oligodendroglioma patients.
*IDH* Isozyme Mutation	Sequencing, Detection of antibody against isozyme	Associated with better prognosis in specific brain tumors, especially diffuse gliomas.
*MGMT* Methylation Status	MSP, qMSP, IHC, pyrosequencing	Indicates a better response to TMZ and radiation therapy, along with improved survival in GBM.
*TERT* Promoter Mutation	Sequencing (Sanger/NGS)	Indicates worse prognosis in primary glioblastomas and oligodendrogliomas.

This table describes multiple biomarkers utilized in detecting and diagnosing brain tumors, their method of detection, and their clinical significance.

## Data Availability

No new data were created or analyzed in this study. Data sharing does not apply to this article.

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
