# Peer review of "Modernizing Neuro-Oncology: The Impact of Imaging, Liquid Biopsies, and AI on Diagnosis and Treatment"

_ijms, 2025, doi:10.3390/ijms26030917_

Round 1

Reviewer 1 Report

Comments and Suggestions for Authors

The manuscript explores cutting-edge topics in neuro-oncology, such as the integration of AI, advanced imaging techniques, and liquid biopsies, making it highly relevant and innovative. The paper is well written with effective technical language, engaging for broad scientific audience. Furthermore, the paper comprehensively reviews challenges and recent advances in the field. The conclusion summarizes promising avenues for research, such as integrating AI with advanced imaging techniques and provides future directions. Nonetheless, following points need to be worked on to enhance the technical aspects:

  1. The focus on AI might in this manuscript masks other innovative aspects, such as metabolic imaging or biomarker integration, which also deserve equal attention. Hence, there should be a balance in all the modalities discussed.
  2. Certain sections, especially the limitations of MRI and the potential of AI, repeat similar points, which could be streamlined.
  3. Quantitative data supporting the advantages of techniques like DTI and radiotracers could be more prominently highlighted to provide evidence-based conclusions.
  4. The figures, for example -regarding AI frameworks (Fig 2) could be better integrated in the corresponding text to enhance readability and contextual understanding.
  5. In general, figures and table can be made clearer and stand-alone by additional annotation or explanation of technical terms for clarity.
  6. A detailed critique of the limitations of the technologies discussed must be included to provide a balanced perspective alongside their advantages.

Reviewer 2 Report

Comments and Suggestions for Authors

The review provides a summary of advancements in neuro-oncology but could be strengthened by providing more evidence, exploring limitations in depth, and enhancing the flow between sections. Below are my specific concerns that need to be addressed.

1) While the review discusses advanced techniques and tools, it could benefit from including specific studies, data, or real-world examples to support claims about their efficacy and impact.

2) The review briefly mentions challenges like accessibility but does not provide an in-depth analysis of potential drawbacks, such as the cost of advanced imaging, ethical considerations in AI, or technical limitations of liquid biopsies.

3) Although informative, the review transitions quickly between topics without deeper exploration. For example, imaging techniques, liquid biopsies, and AI applications are introduced in succession without connecting their interrelated roles in diagnosis and management. Also, the authors should discuss using transformer-based AI models (e.g., PMID: 37798249) in disease in diagnosis.

4) The review advocates for standardization but does not elaborate on the barriers to achieving it, such as variations in equipment, protocols, or regulatory hurdles across institutions.

5) The language is generally clear but could be more concise. For instance, phrases like “designed to address current diagnostic challenges” could be simplified to “addressing diagnostic challenges.”

Round 2

Reviewer 2 Report

Comments and Suggestions for Authors

No more concerns.

Comments on the Quality of English Language

The writing should be polished and improved.

Author Response

Thank you for your feedback regarding the manuscript's writing. We note your comment that the writing should be "polished and improved," and we have taken the following steps to address this concern:

  1. Thorough Editing: The manuscript has undergone multiple rounds of proofreading using advanced tools, including Grammarly and Microsoft Word Editor. Over 100 small revisions were made to enhance clarity, grammar, and style.

  2. Consistency Checks: The text adheres to the International Journal of Molecular Sciences (IJMS) guidelines, with consistency maintained in terminology, grammar, and formatting throughout.

  3. Clarity and Readability: We have ensured the manuscript balances technical rigor with readability by refining complex sentences where necessary without compromising the paper's scientific depth. Given the highly technical nature of this work, we believe the writing appropriately communicates the content to the target academic audience.

We have made every effort to ensure the manuscript meets the highest standards of academic writing. We hope these revisions adequately address your comment.